# Combining organic and mineral fertilizers as a climate-smart integrated soil fertility management practice in sub-Saharan Africa: A meta-analysis

Gil Gram [1,2]*, Dries Roobroeck [3], Pieter Pypers[3], Johan Six[4], Roel Merckx[2], Bernard Vanlauwe[3]

1 Climate Change, Agriculture, and Food Security (CCAFS), International Institute of Tropical Agriculture (IITA), Kampala, Uganda, 2 Department of Earth and Environmental Sciences, KU Leuven, Leuven, Belgium, 3 Climate Change, Agriculture, and Food Security (CCAFS), International Institute of Tropical Agriculture (IITA), Nairobi, Kenya, 4 Department of Environmental Systems Science, Swiss Federal Institute of Technology (ETH), Zürich, Switzerland

* gilgram.fm@gmail.com

**Data Availability Statement:** The meta data file used for the meta-analysis is available from the

## Abstract

Low productivity and climate change require climate-smart agriculture (CSA) for sub-Saharan Africa (SSA), through (i) sustainably increasing crop productivity, (ii) enhancing the resilience of agricultural systems, and (iii) offsetting greenhouse gas emissions. We conducted a meta-analysis on experimental data to evaluate the contributions of combining organic and mineral nitrogen (N) applications to the three pillars of CSA for maize (*Zea mays*). Linear mixed effect modeling was carried out for; (i) grain productivity and agronomic efficiency of N (AE) inputs, (ii) inter-seasonal yield variability, and (iii) changes in soil organic carbon (SOC) content, while accounting for the quality of organic amendments and total N rates. Results showed that combined application of mineral and organic fertilizers leads to greater responses in productivity and AE as compared to sole applications when more than 100 kg N ha$^{-1}$ is used with high-quality organic matter. For yield variability and SOC, no significant interactions were found when combining mineral and organic fertilizers. The variability of maize yields in soils amended with high-quality organic matter, except manure, was equal or smaller than for sole mineral fertilizer. Increases of SOC were only significant for organic inputs, and more pronounced for high-quality resources. For example, at a total N rate of 150 kg N ha$^{-1}$ season$^{-1}$, combining mineral fertilizer with the highest quality organic resources (50:50) increased AE by 20% and reduced SOC losses by 18% over 7 growing seasons as compared to sole mineral fertilizer. We conclude that combining organic and mineral N fertilizers can have significant positive effects on productivity and AE, but only improves the other two CSA pillars yield variability and SOC depending on organic resource input and quality. The findings of our meta-analysis help to tailor a climate smart integrated soil fertility management in SSA.

IITA database (accessible via https://doi.org/10.25502/r259-vt34/d).

**Funding:** The study was funded under the "Integrated soil fertility management for climate smart intensification of maize-based cropping systems in Kenya" project (number PJ-002705) by the Swiss Federal Institute of Technology Zurich (ETH Zurich). The funders had no role in study design, data collection and analysis, decision to publish, or preparation of the manuscript.

**Competing interests:** The authors have declared that no competing interests exist.

## Introduction

Management of fertilizer inputs is of key importance for food security in sub-Saharan Africa (SSA), as its rapidly growing and urbanizing population is not being met with a proportionate growth of agricultural production [1]. Smallholder farmers are suffering from nutrient limitations and low nutrient use efficiencies of particularly nitrogen (N) as a consequence of soil degradation [2–5]. The latter arises when the recycling of crop residues and use of mineral and organic fertilizers are insufficient to compensate for harvested nutrients and soil organic matter losses [1, 6]. So far, most increases in food production in SSA have been achieved by agricultural expansion rather than intensification, which comes at the cost of natural lands and ecosystems. Enhancing the productivity and efficiency of food production in SSA by sustainable intensification is therefore crucial to improve its food security [1, 7].

In addition to soil degradation, climate change is further challenging the agriculture-based societies of SSA [8, 9]. Rainfall is becoming increasingly variable and extreme, and most parts of Africa are experiencing more frequent droughts as observed from long-term precipitation and evaporation trends [10]. Yet, the vast majority of staple food production in SSA, such as maize, comes from rainfed agriculture and is thus vulnerable to changing rainfall regimes [11–14]. Maize is the primary staple food of more than 300 million people in SSA and an important source of income for particularly smallholder farmers [5]. Yet, maize productivity in SSA is low and is being exacerbated by climate change [2, 15, 16]. Maize grain yields on average are 58% below the water-limited potential across tropical Africa [17], and are projected to decline by up to 12% by 2050 [15, 18]. Furthermore, agriculture in SSA is dominated by smallholder subsistence farmers, whose adaptive capacity is low and already being compromised by natural resource degradation [13, 15, 18].

Low productivity and climate change impacts require climate-smart agriculture (CSA), through (i) improving food security by sustainably increasing crop productivity, (ii) enhancing the resilience of agricultural systems or adaptive capacity, and (iii) offsetting greenhouse gas emissions [16, 19–21]. The integrated soil fertility management (ISFM) framework could play an important role in achieving sustainable intensification in SSA. It aims to increase crop productivity by maximizing the agronomic efficiency (AE) of fertilizer inputs through the combined application of mineral and organic fertilizers, improved germplasm, and good agronomic practices [1, 22]. A major component of ISFM is the combined application of mineral and organic fertilizers. Research in SSA has shown that the combined application can lead to greater AE of N and crop productivity for maize as compared to separate applications [23–27]. Several reasons have been suggested. First, combining mineral and organic nutrient sources allows smallholder farmers to apply adequate and proportionate amounts of both minor and major nutrients, which is necessary to sustain soil fertility and crop production in the long term [1, 22]. Second, on top of its obvious role for soil fertility, an increased soil organic matter content also improves other soil functions such as soil biological processes and soil moisture regime [22]. This, in turn, improves resilience to droughts. And third, combined application has the potential to generate interactive effects between both resources, as the synchronization of N availability and plant uptake, both in quantity and time, may be improved through decomposition and subsequent N (im)mobilization processes [25, 28, 29]. Since N synchrony depends on N mineralization and immobilization processes, Vanlauwe et al. [28] hypothesized that organic input quality likely affects potential interactive effects. This was confirmed by incubation and field trials, where Gentile et al. [24, 25] found that low-quality organic inputs tend to favor positive interactions when combined with mineral fertilizer, whereas high-quality organic inputs did not. They argued that the rapid mineralization of high-quality organic N combined with mineral N would exceed early crop demand, and hence would contribute to

potential negative interactions. Similarly, high N input rates could exceed crop demand and lead to negative interactions, suggesting that N input rates may also have an effect on interactive effects [23, 29].

Apart from the effect of combined application on productivity and AE, this study further aims to evaluate the effects on the other two pillars of CSA, namely the agricultural system's resilience and offsetting of greenhouse gas emissions such as $CO_2$. Concerning the resilience of food production systems, yield reliability can be considered as a key indicator as it defines the capacity of the system to remain close to its stable yield equilibrium when exposed to seasonal variation in weather conditions [30]. Since climate change is rendering the latter more variable and extreme, yield variability is expected to increase, and as such will decrease yield reliability [31, 32]. Yet, despite its importance, research addressing maize yield variability or reliability in the context of nutrient trials is scarce. Nevertheless, Bayu et al. [33] found through sorghum trials in Ethiopia that yield stability decreased with the applications of manure combined with higher rates of mineral N, Vanlauwe et al. [34] concluded that combining mineral N with alley-cropping *Senna siamea* in maize-cowpea rotations would outperform the other treatments while achieving an acceptable yield stability, and Fujisaki et al. [35] stated that high organic inputs are needed in the tropics to ensure yield stability and climate change resilience.

Emissions of $CO_2$, on the other hand, can potentially be mitigated by storing atmospheric carbon (C) in soils. Apart from the mitigation potential, raising soil organic carbon (SOC) stocks is also essential for a sustainable soil health and crop productivity [36, 37]. Raising SOC stocks seems, however, to be challenging. Long-term organic and mineral nutrient trials on maize monocropping in SSA consistently reported declining SOC [38–42], but found that treatments with organic inputs were more successful in reducing SOC losses than mineral and no-input treatments [40, 41, 43–45]. A meta-analysis conducted by Chivenge et al. [23] confirmed that mineral fertilizer treatments did not have a significant effect and that organic inputs were necessary to help reduce SOC loss. Whether organic matter inputs contribute to stable soil organic matter or SOC, depends on a number of factors influencing the decomposition and subsequent stabilization processes, such as organic input properties, and soil and climate variables [46–48]. Fujisaki et al. [35] found that organic input quantity and quality were more important predictors than soil and climate properties, though quantity likely has a larger effect than quality according to Gentile et al. [29] and Castellano et al. [49]. In addition, Chivenge et al. [46] and Gentile et al. [50] did not observe quality related long-term differences in SOC stocks.

With climate change adaptation and mitigation of food production systems becoming ever more critical, the need arises for a complete CSA assessment of the ISFM combined application practice. We therefore conducted a meta-analysis of short- and long-term maize nutrient trials across SSA, in order to assess the effect of combined versus sole application of organic and mineral N inputs on (i) maize productivity and AE of N, (ii) maize yield variability, and (iii) on SOC. In parallel, we investigated to what extent these effects were influenced by organic matter quality and total N input rates.

## Materials and methods

### Data collection

Relevant peer-reviewed publications were identified through the following key word searches on Google Scholar: 'combined application', 'Sub-Saharan Africa', 'maize', 'organic', and 'mineral'. Studies were selected when providing data on maize grain yields and associated variance for treatments with (i) sole application of mineral N fertilizer (MR), (ii) sole application of unprocessed organic resources (OR), and (iii) combined application of mineral and organic

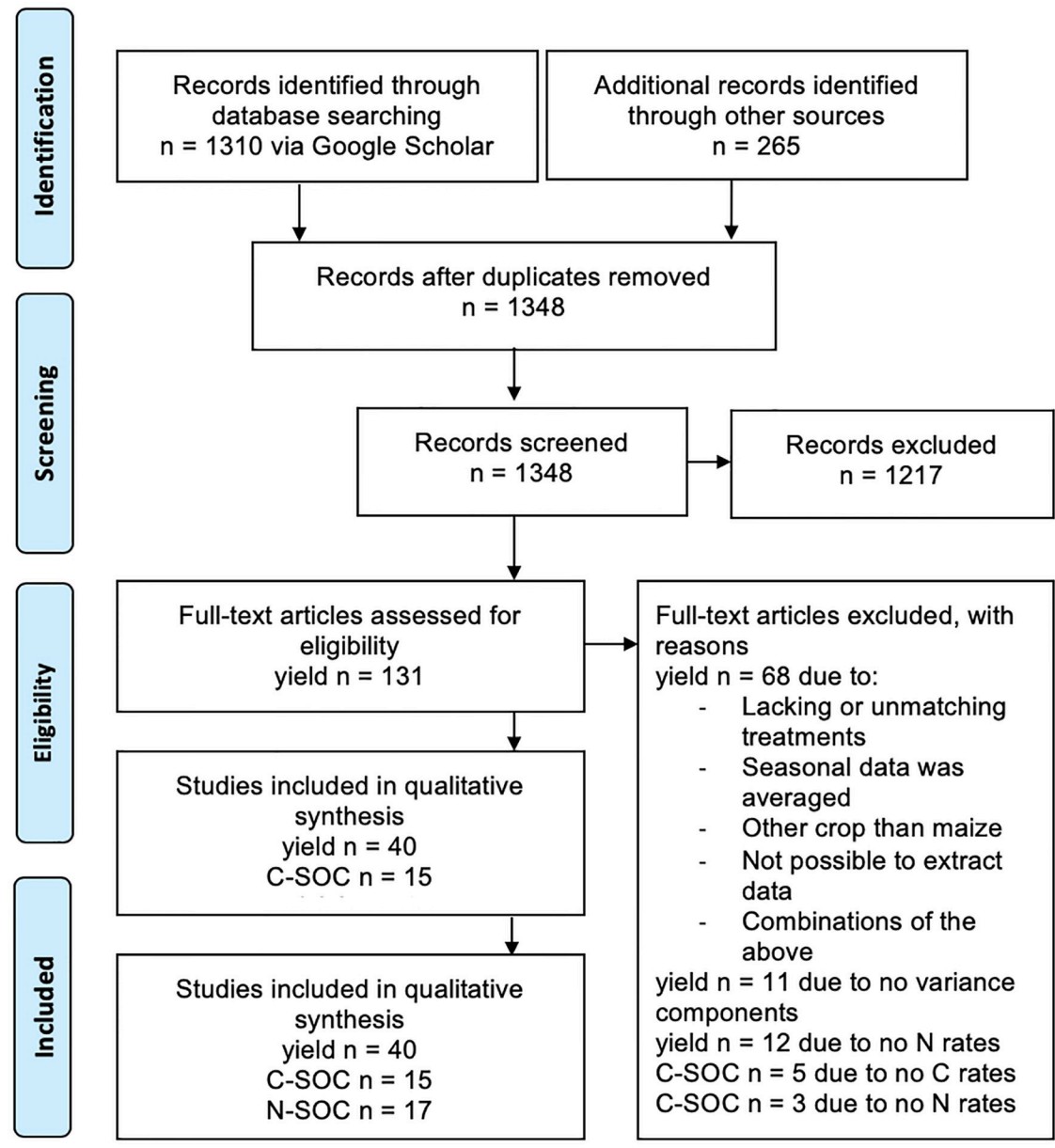

**Fig 1. Preferred Reporting Items for Systematic Reviews and Meta-Analysis (PRISMA) Flow Chart describing the protocol used for searching, identifying, and selecting publications for the current meta-analysis.** 'n' represents the total number of studies, or for the specific analyses if specified (yield, C-SOC, and N-SOC).

resources (ORMR). This first selection process refers to steps one to three of the PRISMA flow chart in Fig 1. A last search was conducted on 9/3/2020, after which no more data were included. After consolidating the data (Cfr. section Data consolidation), the selection of studies was further refined by screening for studies with organic resource type and quality parameters (N, C, polyphenols, lignin content, and C:N ratio), organic and mineral input rates for the productivity and AE analyses, and initial and measured SOC with organic C and N rates for the SOC analyses with C and N inputs, respectively. This second selection round refers to steps four to six of the Preferred Reporting Items for Systematic Reviews and Meta-Analyses

(PRISMA) flow chart in Fig 1. By applying the abovementioned selection criteria, 40 studies were used in total for the productivity and variability analysis, providing 2943 yield data points across 68 sites. Of those 40 studies, 15 provided 488 SOC data points across 21 sites with C rate data, and 17 provided 531 SOC data points across 23 sites with organic N rate data. An overview of the relevant data from the publications selected for each analysis is given in Table 1.

A no-input treatment (control) was not required, but was absent in only two studies, i.e. Mungai et al. [54] and Nziguheba et al. [83]. Studies with fallow rotations were excluded as residual effects were not part of the research questions. When the data were not readily available from tables, they were extracted from graphs using GraphClick [91], or an attempt was made to request the authors for their data. Raw data from publications were obtained from Vanlauwe et al. [92], Franke et al. [93], and the meta-analysis of Chivenge et al. [23], and may have included gray literature data. Unpublished data were obtained from the works of Chivenge et al. [90], Fonte et al. [81], and Mapfumo et al. [89]. For different publications covering the same trial, only the most exhaustive one was considered (e.g. Vanlauwe et al. [34, 92]). The following data were also retrieved from all papers: country and site location (name and GPS coordinates), growing season or year, maize grain yield and variance, type of organic application and cropping system, and mineral fertilizer design.

## Data consolidation

Information from publications was compiled for each individual site within study. A standardized time measure was assigned to all observations, corresponding to the $n^{th}$ growing season of the respective experiment. This could be one or two per year, depending on uni- or bimodal rainfall patterns, respectively. Studies were classified according to their type of organic application or cropping system, i.e. monocrop (with biomass inputs), legume intercrop, and legume rotation. Mineral fertilizer design indicated if phosphorus (P) and potassium (K) are either not applied (fixed), applied as a blanket for all treatments including a control (fixed), or applied proportionally with N (NPK). The study of Akinnifesi et al. [43] concerns three different P rates, so it is used once as an NPK study (3 different MR treatments), and another three times as blanket studies (one for each P rate). Organic resource types included raw residues from crops, biomass from alley crops, legume rotation or intercropping, and animal manure. These were categorized into four quality classes according to the organic data base (ODB) of Palm et al. [47]. High-quality organic classes I and II have a high N content (>2.5%), and low-quality classes III and IV have a low N content (<2.5%). Classes I and III have low lignin contents (<15%), whereas classes II and IV have high lignin contents (>15%) (Table 2). A separate class Manure was created to include variable organic inputs like farm yard manure and compost, analogous to Vanlauwe et al. [94]. When not reported, quality parameters were estimated using average values from the ODB, or otherwise other literature available. Only a few studies addressed organic resources of class IV, and for a limited range of N inputs. As a consequence, class IV was considered too unbalanced throughout the data set and was not used for the yield and AE modeling. It could, however, be included in the SOC modeling with C input as it was balanced across the C input range. The combinations of treatment with the organic classes are hereafter referred to as subtreatments. When organic input rates were reported as tons of biomass added per ha, the organic N and C % input was estimated using the ODB of Palm et al. [47]. No distinction was made for the different experimental designs or maize varieties, nor were different planting stands and densities adjusted for. Yield data were used as they were reported from studies, i.e. as averages from replicated treatments. Standard deviations (SDs) were conservatively derived from the variance components available, for each reported yield estimate if possible. The SOC data of a publication were considered relevant when at least two

**Table 1. Overview of the data set used for the YIELD and SOC meta-analyses, with columns for the study*country*site combination index (id), study (reference), aridity index class (AI class)*, time span of the study's experiment (time span), number of yield data (nYield), number of SOC data associated with C and N rates (nSOC C input and nSOC N input, respectively) (*) The site's aridity index was extracted from the CGIAR-CSI Global-Aridity and Global-PET Database [51].**

| id | reference | country | site | AI class | time span | cropping system | fertilizer design | nYield | nSOC C input | nSOC N input |
|---|---|---|---|---|---|---|---|---|---|---|
| 9 | Mariki et al. [52] | Tanzania | Selian arusha | humid | 2 | monocrop | fixed | 10 | | 5 |
| 11 | Kihanda et al. [53] | Kenya | Embu Kihanda | dry sub-humid | 10 | monocrop | NPK | 24 | 16 | 16 |
| 12 | Kihanda et al. [53] | Kenya | Kavutiri | humid | 9 | monocrop | NPK | 8 | 8 | 8 |
| 21 | Mungai et al. [54] | Kenya | Njoro Makuru | dry sub-humid | 1 | monocrop | fixed | 10 | | |
| 22 | Mungai et al. [54] | Kenya | Piave Makuru | dry sub-humid | 1 | monocrop | fixed | 10 | | |
| 25 | Murwira et al. [55] | Zimbabwe | Muchinjike | dry sub-humid | 1 | monocrop | fixed | 12 | | |
| 28 | Murwira et al. [55] | Zimbabwe | Wedza | dry sub-humid | 1 | monocrop | fixed | 12 | | |
| 29 | Onyango et al. [56] | Kenya | Anin Kitale | dry sub-humid | 3 | monocrop | fixed | 18 | | |
| 30 | Onyango et al. [56] | Kenya | Cheptya kitale | dry sub-humid | 2 | monocrop | fixed | 8 | | |
| 31 | Onyango et al. [56] | Kenya | Chobosta kitale | dry sub-humid | 3 | monocrop | fixed | 12 | | |
| 32 | Onyango et al. [56] | Kenya | Matunda Kitale | dry sub-humid | 3 | monocrop | fixed | 12 | | |
| 33 | Snapp [57] | Zimbabwe | Domboshawa RS | dry sub-humid | 1 | rotation | NPK | 4 | | |
| 34 | Workayehu [58] | Ethiopia | Awasa | dry sub-humid | 3 | monocrop | fixed | 54 | | |
| 35 | Ikerra et al. [59] | Malawi | Makoka RS | humid | 2 | intercrop | fixed | 12 | | |
| 36 | Kapkiyai et al. [40] | Kenya | Kabete RS | humid | 19 | monocrop | NPK | | 4 | |
| 37 | Mugendi et al. [60] | Kenya | Embu RS | humid | 34 | monocrop | fixed | | 6 | 6 |
| 40 | Makumba et al. [61] | Malawi | Makoka RS | humid | 2 | monocrop | fixed | 12 | | |
| 41 | Mugendi et al. in Okalebo et al. [62, 63] | Kenya | Meru | humid | 2 | monocrop | fixed | 28 | | |
| 42 | Nandwa [64] | Kenya | Kabete | humid | 5 | monocrop | NPK | | | 4 |
| 43 | Gigou and Bredoumy [65] | Ivory Coast | Gagnoa | humid | 20 | monocrop | fixed | 120 | | 30 |
| 44 | Iwuafor et al. [66] | Benin | Derived Savannah | dry sub-humid | 2 | monocrop | fixed | 8 | | |
| 45 | Iwuafor et al. [66] | Nigeria | N-Guinea savannah | humid | 2 | monocrop | fixed | 8 | | |
| 48 | Nhamo [67] | Zimbabwe | Chimombe Mrewa2 | dry sub-humid | 1 | monocrop | fixed | 6 | | |
| 49 | Nhamo [67] | Zimbabwe | Chinonda Mrewa3 | dry sub-humid | 1 | monocrop | fixed | 6 | | |
| 50 | Nhamo [67] | Zimbabwe | Chisunga Mrewa2 | dry sub-humid | 1 | monocrop | fixed | 6 | | |
| 51 | Nhamo [67] | Zimbabwe | Kaitano Mrewa2 | dry sub-humid | 1 | monocrop | fixed | 6 | | |
| 52 | Nhamo [67] | Zimbabwe | Mangena Mrewa2 | dry sub-humid | 1 | monocrop | fixed | 6 | | |
| 53 | Nhamo [67] | Zimbabwe | Manjoro Mrewa2 | dry sub-humid | 1 | monocrop | fixed | 6 | | |
| 54 | Nhamo [67] | Zimbabwe | Mapira Mrewa2 | dry sub-humid | 1 | monocrop | fixed | 6 | | |

(*Continued*)

**Table 1.** (Continued)

| id | reference | country | site | AI class | time span | cropping system | fertilizer design | nYield | nSOC C input | nSOC N input |
|----|-----------|---------|------|----------|-----------|-----------------|-------------------|--------|--------------|--------------|
| 55 | Nhamo [67] | Zimbabwe | Mukudu Mrewa | dry sub-humid | 1 | monocrop | fixed | 6 | | |
| 56 | Nhamo [67] | Zimbabwe | Sunha Mrewa2 | dry sub-humid | 1 | monocrop | fixed | 6 | | |
| 57 | Nhamo [67] | Zimbabwe | Zivhu Mrewa | dry sub-humid | 1 | monocrop | fixed | 6 | | |
| 58 | Nziguheba et al. [27] | Kenya | Nyabeda | humid | 2 | monocrop | NPK | 14 | | |
| 59 | Uyovbisere and Elemo [68] | Nigeria | Samaru RS | dry sub-humid | 4 | monocrop | NPK | 24 | 16 | 16 |
| 63 | Mucheru-Muna et al. [69] | Kenya | Chuka off station | humid | 4 | monocrop | fixed | 40 | | |
| 64 | Mucheru-Muna et al. [69] | Kenya | Chuka on-farm | humid | 3 | monocrop | fixed | 22 | | |
| 65 | Nyamangara et al. [70] | Zimbabwe | Domboshawa RS | dry sub-humid | 3 | monocrop | fixed | 18 | | |
| 67 | Chilimba et al. in Munthali [71, 72] | Malawi | Bvumbwe | humid | 1 | monocrop | fixed | 14 | | |
| 68 | Chilimba et al. in Munthali [71, 72] | Malawi | Chitedze RS | dry sub-humid | 1 | monocrop | fixed | 14 | | |
| 69 | Delve [73] | unknown | unknown | | 2 | monocrop | fixed | 7 | | |
| 70 | Kimani et al. [74] | Kenya | Kariti | humid | 2 | monocrop | NPK | 20 | | |
| 71 | Kimani et al. [74] | Kenya | Gatuanyaga | semi-arid | 1 | monocrop | NPK | 10 | | |
| 72 | Kimetu et al. [75] | Kenya | Kabete RS | humid | 1 | monocrop | fixed | 8 | | |
| 73 | Okalebo et al. [76] | Kenya | Eldoret | dry sub-humid | 4 | monocrop | fixed | 18 | | |
| 74 | Ayoola and Adeniyan [77] | Nigeria | Oniyo | humid | 2 | monocrop | NPK | 8 | | |
| 75 | Ayoola and Adeniyan [77] | Nigeria | Moloko-ashipa | humid | 2 | monocrop | NPK | 8 | | |
| 77 | Akinnifesi et al. [43] | Malawi | Makoka RS | humid | 15 | intercrop | fixed | 136 | 24 | 24 |
| 77 | Akinnifesi et al. [43] | Malawi | Makoka RS | humid | 15 | intercrop | NPK | 136 | 22 | 22 |
| 78 | Kimani et al. [78] | Kenya | Githunguri kiambu | humid | 1 | monocrop | fixed | 8 | | |
| 79 | Kimani et al. [78] | Kenya | Kariti Maragwa | humid | 1 | monocrop | fixed | 8 | | |
| 80 | Kimani et al. [78] | Kenya | Mukanduini Kirinyaga | humid | 1 | monocrop | fixed | 8 | | |
| 81 | Mugendi et al. [79] | Kenya | Embu RS | humid | 22 | monocrop | fixed | 108 | | |
| 82 | Mugwe et al. [80] | Kenya | Chuka on-farm 2 | humid | 4 | monocrop | fixed | 63 | | |
| 82 | Mugwe et al. [80] | Kenya | Chuka on-farm 2 | humid | 4 | intercrop | fixed | 63 | | |
| 83 | Mugwe et al. [80] | Kenya | Chuka on-farm 3 | humid | 4 | monocrop | fixed | 33 | | |
| 83 | Mugwe et al. [80] | Kenya | Chuka on-farm 3 | humid | 4 | intercrop | fixed | 33 | | |
| 84 | Mugwe et al. [80] | Kenya | Kirege school Chuka | humid | 4 | monocrop | fixed | 64 | | |
| 84 | Mugwe et al. [80] | Kenya | Kirege school Chuka | humid | 4 | intercrop | fixed | 64 | | |
| 90 | Fonte et al. [81] | Ghana | Kwadaso RS | humid | 7 | monocrop | fixed | 70 | 22 | 22 |
| 92 | Kimaro et al. [82] | Tanzania | Dodoma2 | semi-arid | 2 | monocrop | fixed | 18 | | |
| 93 | Nziguheba et al. [83] | Nigeria | Samaru Zaria | dry sub-humid | 10 | monocrop | fixed | 90 | | |
| 94 | Nziguheba et al. [83] | Benin | Sekou | humid | 10 | monocrop | fixed | 90 | 18 | 18 |
| 94 | Nziguheba et al. [83] | Benin | Sekou | humid | 10 | intercrop | fixed | | 6 | 6 |
| 95 | Shisanya et al. [84] | Kenya | Kirege | humid | 5 | monocrop | NPK | 52 | 12 | 14 |
| 96 | Anyanzwa et al. [85] | Kenya | Teso | humid | 2 | monocrop | fixed | 16 | 8 | 8 |

(*Continued*)

**Table 1.** (Continued)

| id | reference | country | site | AI class | time span | cropping system | fertilizer design | nYield | nSOC C input | nSOC N input |
|---|---|---|---|---|---|---|---|---|---|---|
| 105 | Vanlauwe et al. [34] | Nigeria | Ibadan IITA campus1 | humid | 21 | monocrop | NPK | 72 | 26 | 26 |
| 106 | Mutegi et al. [86] | Kenya | Mucwa | humid | 2 | monocrop | fixed | 20 | 8 | 10 |
| 107 | Bedada et al. [45] | Ethiopia | Beseku | dry sub-humid | 6 | monocrop | NPK | 20 | 4 | 4 |
| 108 | Mucheru-Muna et al. [39] | Kenya | Mucwa poor | humid | 7 | monocrop | fixed | 70 | 8 | 10 |
| 109 | Mucheru-Muna et al. [39] | Kenya | Mucwa rich | humid | 7 | monocrop | fixed | 70 | 7 | 9 |
| 110 | Detchinli and Sogbedji [87] | Togo | Lome RS | dry sub-humid | 4 | monocrop | NPK | 16 | | |
| 111 | Tovihoudji et al. [88] | Benin | CRA-Nord | dry sub-humid | 4 | monocrop | NPK | 30 | 21 | 21 |
| 116 | Mapfumo et al. [89] | Zimbabwe | Domboshawa SOM | semi-arid | 6 | monocrop | fixed | 90 | | |
| 117 | Mapfumo et al. [89] | Zimbabwe | Makoholi SOM | dry sub-humid | 5 | monocrop | fixed | 71 | | |
| 118 | Chivenge et al. [90] | Kenya | Embu RS block B | dry sub-humid | 32 | monocrop | fixed | 270 | 98 | 98 |
| 119 | Chivenge et al. [90] | Kenya | Machanga | semi-arid | 33 | monocrop | fixed | 243 | 88 | 88 |
| 120 | Fonte et al. [81] | Ghana | Ayuom | humid | 9 | monocrop | fixed | 72 | 22 | 22 |
| 121 | Chivenge et al. [90] | Kenya | Aludeka | humid | 27 | monocrop | fixed | 234 | 22 | 22 |
| 122 | Chivenge et al. [90] | Kenya | Sidada | humid | 27 | monocrop | fixed | 234 | 22 | 22 |

measurements over time were reported, so that responses could be calculated by subtracting the initial SOC from the SOC at a given time during or after the experiment (dSOC). Finally, soil and climate data were not sufficiently reported and therefore not included in the analysis. See Table 1 for a complete overview of the data.

**Table 2. List of organic resource types and associated quality classes, with columns for the respective number of studies (nReferences), number of yield data (nYield), and number of SOC data with C and N rates different from zero (nSOC C input and N input, respectively).**

| type | class | nReferences | nYield | nSOC C input | nSOC N input |
|---|---|---|---|---|---|
| *Crotalaria juncea* (Sun hemp) | I | 3 | 74 | 10 | 10 |
| *Crotalaria ochroleuca* | I | 2 | 23 | | |
| *Gliricidia sepium* | I | 3 | 84 | 24 | 24 |
| *Glycine max* (Soybean) | I | 1 | 6 | | |
| *Parkia biglobosa* (Locust bean) | I | 1 | 9 | 6 | 6 |
| *Tithonia diversifolia* | I | 11 | 316 | 50 | 50 |
| *Azadirachta indica* (Neem tree) | II | 2 | 89 | 14 | 14 |
| *Calliandra calothyrsus* | II | 10 | 365 | 51 | 51 |
| *Leucaena leucocephala* | II | 7 | 135 | 20 | 20 |
| *Mucuna pruriens* | II | 4 | 57 | 6 | 6 |
| *Senna siamea* (Cassia tree) | II | 3 | 106 | 16 | 16 |
| *Arachis hypogaea* (Groundnut) | III | 1 | 2 | | |
| Coffee | III | 1 | 48 | | |
| Maize residue | III | 7 | 307 | 52 | 55 |
| *Triticum aestivum* (Wheat) | III | 1 | 6 | | |
| Sawdust | IV | 2 | | 45 | 45 |
| Compost | Manure | 4 | 88 | 2 | 17 |
| Farm yard manure | Manure | 22 | 579 | 81 | 89 |

## Statistical analysis

All statistical computing and graphic design for this paper was carried out in R [95](version 3.6.1). Yield data were modeled using the multivariate meta-analytical mixed effects function *rma.mv* of the Metafor package, which takes the sampling variances into account [96]. The advantage is that this function allows to account for yield variation and covariance among treatments, thereby avoiding biased estimates, overestimating variances, and increasing chances of finding false significant effects [97, 98]. For the SOC data, on the other hand, the reporting of sampling variances was too scarce, hence the lmer function of the lme4 package was used [99]. As opposed to *rma.mv*, the *lmer*function assumes sampling variances are not exactly known. The first is thus essentially a special case of the latter [96]. Absolute yield and SOC values are used for analysis instead of effect sizes. As such, the model can account for variation in yields and SOC of control treatments and their covariance with MR, OR and MROR.

**Yield analysis.** The linear mixed effect model for the yield analysis (YIELD) was composed of the square root transformed absolute yield as the outcome variable, together with fixed and random effect terms. Residual and QQ plots revealed no significant issues for linearity and normality, but that a square root transformation was necessary to improve homoscedasticity. The choice of fixed effects, also called predictors or moderators, is hypothesis-based and expresses the particular interest in each of their level's effect on productivity and variability. Maximum likelihood tests were used to compare models with and without fixed effects, in order to evaluate their contribution to the model. Random effects, on the other hand, are chosen based on their significant contribution to explained variance, while aiming for the most parsimonious model. Their contribution's significance was evaluated using restricted maximum likelihood tests. As such, fixed effects were included in the model for the following variables: mineral N input rates, organic N input rates of each organic class, the interactions between them, the quadratic term of each rate, and also their quadratic terms. Furthermore, fixed effects were included for cropping system and mineral fertilizer design to correct for differences in overall mean yield. As for the random effect structure, random intercepts were included for study, individual observations, site, and time nested within site (hereafter referred to as time). In order to also account for the correlated random effects of the different treatments within site and within time, treatment is included in the random structure as random slopes for the random intercepts site and time. The random terms allow the model to account for their inherent differences and non-independences.

In this study, interactive effects are defined as the difference in yield response when applied in combination, versus the sum of the yields when applied solely as OR and MR. They are quantified according to Eq 1 [28]:

$$\text{IE} = Y_{ORMR} - Y_{control} - \left( Y_{MR} - Y_{control} \right) - \left( Y_{OR} - Y_{control} \right) \tag{1}$$

where IE means the interactive effect and $Y_{control}$, $Y_{MR}$, $Y_{OR}$, and $Y_{ORMR}$ are the mean yield estimates of the respective Control, MR, OR, and ORMR treatments. A positive interaction is thus observed when the combined application (with 50% OR and 50% MR) yields more than the sum of the sole applications. This means that a yield response curve of ORMR with a positive interaction would sit above the ORMR dashed line (b) as illustrated in Fig 2 below.

The coefficients of the YIELD model were used to predict yield estimates and construct mean yield response and AE curves by total N input rate. Predictions were done for total N input rates for which there were actual yield observations, but up until a maximal total N input rate of 200 kg N ha$^{-1}$, after which inputs are considered not sensible anymore. These predictions were also done for a fixed mineral fertilizer design and separately for each organic class and ORMR treatments with three different organic N proportions, i.e. 25, 50, and 75% (Cfr.

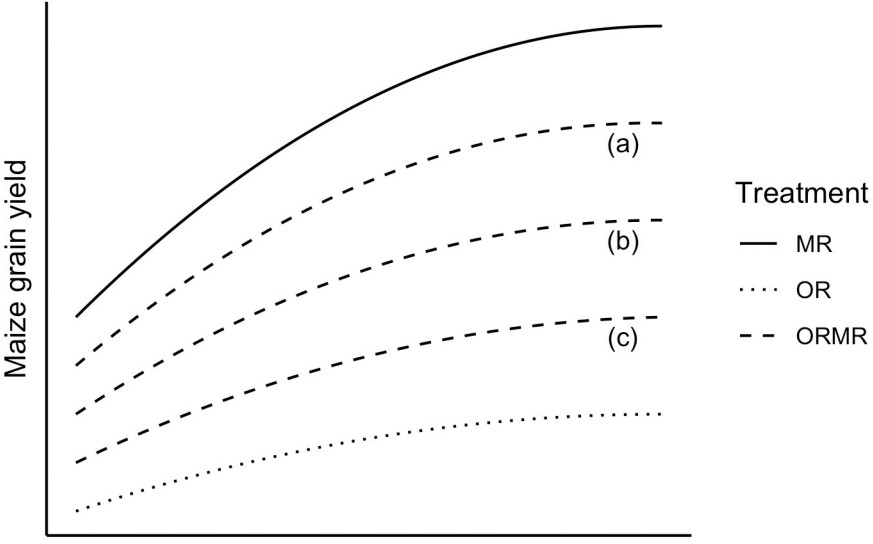

**Fig 2. Graph illustrating conceptual response curves of a mineral, organic, and combined input treatment with substitutive design (50% mineral, 50% organic) with interactive effects that can be (a) positive, (b) none existent, and (c) negative.**

section Modeled yields and agronomic efficiency). The AE was computed in accordance to Vanlauwe et al. [94], i.e. by taking the differences in predicted maize grain yields of MR, OR and ORMR from the Control (kg ha$^{-1}$), and dividing these by the amount of N applied (kg kg$^{-1}$).

The random variance, as modeled by the YIELD model's random structure, can be used as an estimation of yield variability for each treatment across time. The interest of this paper, however, does not reside in the variability of each treatment, but in the variability of each sub-treatment instead. Fitting random slopes for subtreatments was not possible, because covariances would have needed to be estimated between subtreatments that do not make sense or for which no data are available. As an alternative, the subtreatment variances were extracted from the random structure of models that were run for subsets of each organic quality class separately.

The YIELD model and individual organic class models therefore provided yield variance and covariance estimations across time. These were then used to compute variance responses using Eq 2:

$$\text{var}_{response} = var_{treatment} + var_{control} - 2 * covar_{treatment,control} \tag{2}$$

with var and covar being the variance and covariance, respectively. The responses were subsequently divided by the mean predicted yields from the YIELD model to obtain relative variance responses. Note that since the data were modeled using a square root transformation, the variance responses are in fact SD responses in t ha$^{-1}$ for the original data.

**SOC analysis.** Two separate linear mixed effect models were designed to model dSOC, one with organic N rates as predictors (N-SOC model) and one with organic C rates (C-SOC model). Stocks of SOC could not be modeled due to a lack of reported soil bulk densities. For the N-SOC model, fixed effects were included for organic N rates of different quality classes, mineral N rate, the quadratic terms and interactive terms of both. For the C-SOC model, fixed effects were included for organic C rates of different quality classes, mineral N rate, and

interactive terms of both. Fitting quadratic terms was not possible. Both models also included fixed effects for initial SOC and mineral fertilizing design for model fitting purposes in analogy with the YIELD model. Cropping system was not relevant, as all SOC reporting studies were monocropping with biomass application. Both models have the same random intercepts for site and time. Residual and QQ plots confirmed that there were no significant violations of linearity, normality, and homoscedasticity and hence no transformations were needed.

The outputs of the N-SOC and C-SOC models are used to predict dSOC across cumulative N and organic C input rates (t ha$^{-1}$) for which data are available, at a data set average initial SOC of 1.8% and for a fixed mineral fertilizer design. For the N-SOC graph these dSOC are plotted separately for each organic class and ORMR with different proportions of organic N input, while for the C-SOC graphs these are plotted for each treatment and organic class at a cumulative mineral N input of 750 kg ha$^{-1}$.

**Publication bias and sensitivity analyses.** Publication bias was assessed by funnel plot symmetry, which was statistically tested by adding the sampling variance as a moderator in the model as a way of extending Egger's test to complex models [100, 101]. While the funnel plot looked slightly asymmetric, the Egger's test confirmed it was significantly symmetric ($p < 0.05$), therefore suggesting the absence of publication bias. A sensitivity analysis was carried out for the YIELD model, by re-fitting the model without the most influential studies. Based on the diagonal values of the Hat matrix, four studies (469 or 17% of total data points) were identified as relatively most influential, while five studies (1478 or 52% of total data points) were identified based on their Cook's distances [100]. Removing these study sets did not change the overall trends and conclusions for the yield analysis, with the exception of organic class II and to a greater extent III that were substantially underrepresented in the smallest Cook's distance subset. The current YIELD model was therefore considered to be sufficiently robust. The SOC model was fitted with a considerably smaller amount of data, and was therefore considered more sensitive to influential studies. In fact, two out of the 15 studies provided long-term data accounting for almost 50% of the total amount of data. Leaving them out did impact the models and consequent findings considerably. However, long-term data on SOC in SSA are scarce, yet could provide valuable insights [102]. We therefore nevertheless consider that including the available long-term data is justified and can contribute to the research questions of this paper, despite their relatively large influence on the results.

## Results

### Overview original data

The frequency distribution of yield data across total N rates gives an indication of the amount of data available by N rate, as shown by the density graph of Fig 3. The total N input rate for the MR treatments ranged from 20 to 175 kg N ha$^{-1}$, for OR from 10 to 524 kg N ha$^{-1}$, and for ORMR from 45 to 645 kg N ha$^{-1}$. The graph suggests a potential bias towards higher yields for ORMR treatments compared to MR and OR, and to a lesser extent MR compared to OR. This implies that N rate is a necessary covariate in the meta-analysis.

Fig 4 shows the average and variation of yields for the different treatments, across all publications and study sites, and for each individually. The graph reveals that on average ORMR treatments tend to represent higher yield averages, followed by MR and OR, respectively. The control treatments represent, as expected, the lowest averages.

With the reported yield data, one can calculate the AE across total N input and by treatment and organic class (Fig 5). From these calculations it is observed that the AE decreases in value as well as in variability with increasing N input. Differences between organic qualities seem to confirm this, as classes with a low N content tend to be more variable and lead to both higher

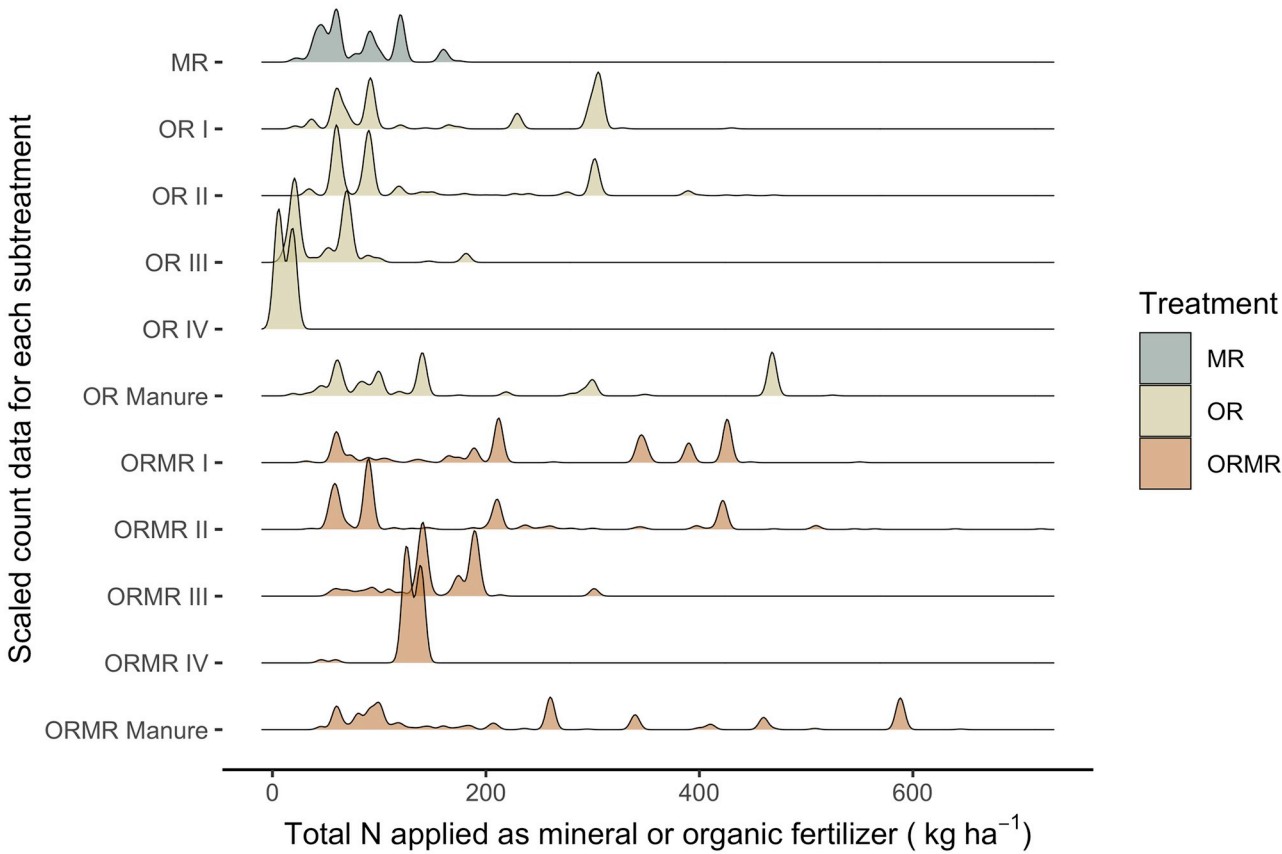

**Fig 3. Distribution of data points, represented by scaled count data for each subtreatment across total N rate, applied as mineral or organic fertilizer.**

and lower AE values than classes with high N content. This illustrates the importance of analyzing organic quality class and N rate with mixed effect models, where the variance between those classes in relation to N rate can be taken into account.

## Modeled yields and agronomic efficiency

The YIELD model results (S1 Table) show significant positive effects for input rates of each of the organic quality classes and mineral N, meaning that their mean maize grain yields are estimated to be significantly greater than the control treatment. The model output also revealed a significant interaction between organic and mineral N rates for all organic classes ($P < 0.05$), meaning that the effect of organic inputs in the combined ORMR treatment depends on the estimated effect of mineral N, and vice versa. Intercropping as a cropping system has a significant yield decline as opposed to monocrop with biomass inputs, whereas rotation is not significantly different from the latter. Mineral fertilizing design does not have a significant effect on yields. The relationship of model predicted yields and their derived AE with total N rate is visualized in Figs 6 and 7, respectively.

Predicted yields increase with increasing total N rate, but only up to a certain point, which depends on treatment and organic quality. The combined application ORMR consistently leads to greater yields than OR across the whole range of total N inputs. Compared to MR, however, its relative effect depends on total N input, organic class, and its proportion of organic N. Below total N rates of 100-125 kg N ha$^{-1}$, the ORMR of all organic classes with 25%

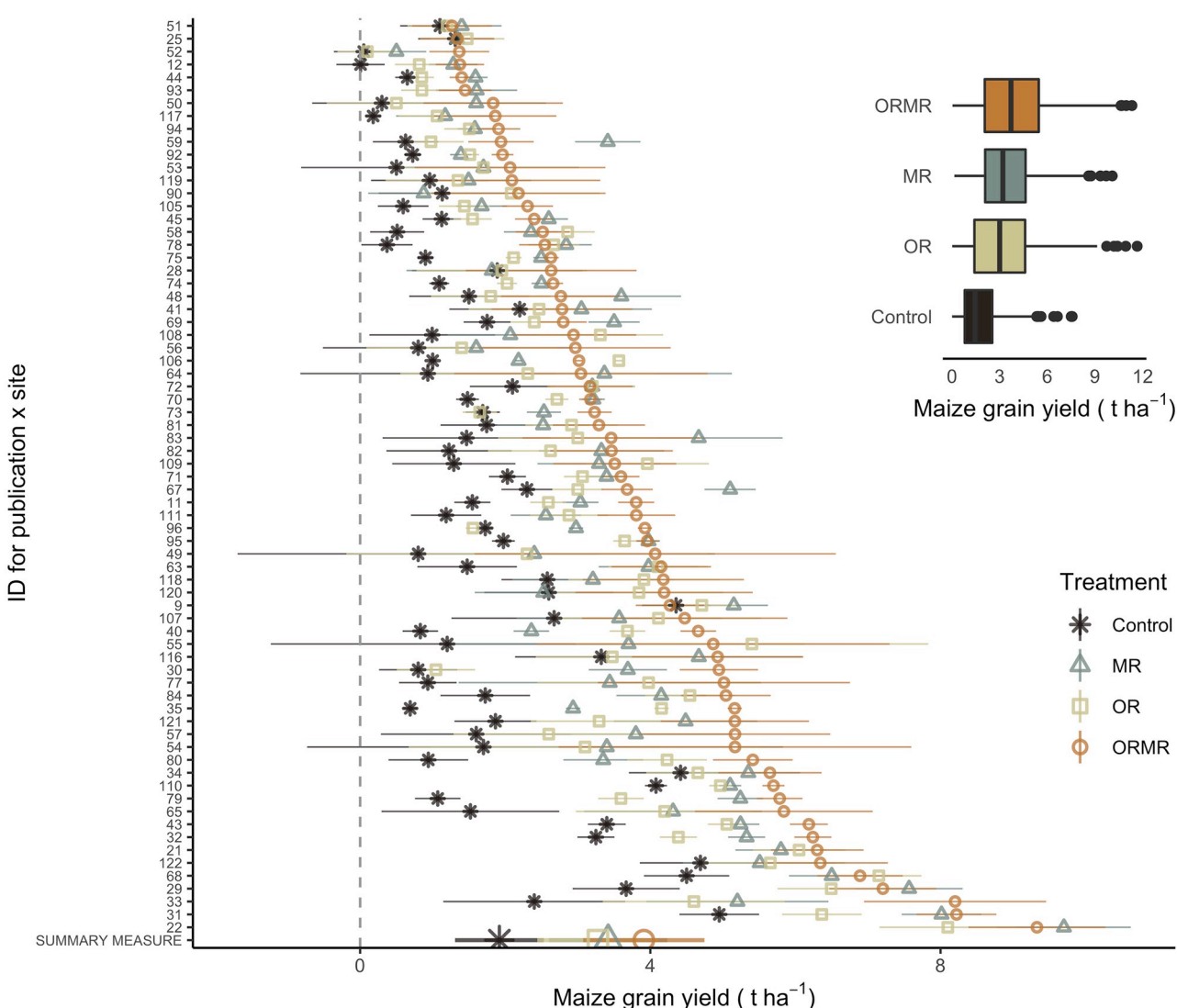

**Fig 4. (left) Overview of reported maize grain yield (t ha$^{-1}$) data, averaged by publication, site, and treatment, and ranked following increasing ORMR yields.** Error bars represent the +- 1.96 * mean standard error on that average. The overall yield average by treatment and across publications and sites is labeled as 'summary measure' at the bottom of the graph. (top right) Distribution of maize grain yield (t ha$^{-1}$) values by treatment.

organic N have similar yields to that of MR. With higher organic N proportions, the predicted ORMR yields tend to gravitate towards the lower yield predictions of OR. These reduced responses get more pronounced in the order of class I, II, Manure, and in particular class III. For the latter class, OR and ORMR with organic N proportions greater than 50% exhibit culminating and decreasing ORMR responses within the observed range of input rates. On the other hand, with total N rates exceeding 100-125 kg total N ha$^{-1}$, the MR response curve flattens, while those of OR and ORMR (except for organic class III) do not. As a result, ORMR will outperform MR at a certain N rate, depending again on organic class and proportion. The 25% organic ORMR of classes I and II outperform MR earlier than classes III and Manure. For ORMR with higher organic proportions, one needs higher N rates to outperform MR. For organic class III, however, only the 25% organic ORMR eventually outperforms MR within the 0-200 kg total N input range.

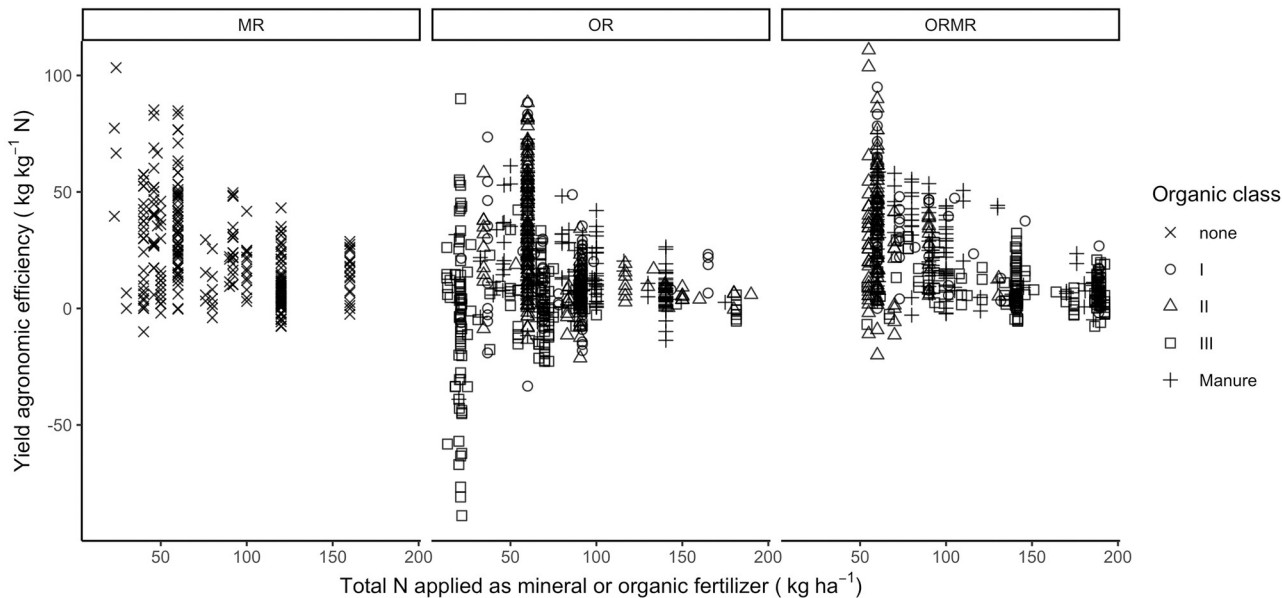

**Fig 5. The agronomic efficiency (AE, in kg kgN⁻¹) across a total N input range of 0-200 (kg ha⁻¹), applied as mineral or organic fertilizer, and segregated by treatment and organic input quality class.**

Apart from OR class III, all subtreatments have a similar decreasing AE with increasing total N input, i.e. the decrease is fastest for MR, slowest for OR and intermediate for ORMR. The inclinations of the latter AE curves increase with more mineral N, and decrease with higher proportions of organic N. Class III is the exception as it is the only organic class with a rapidly declining AE for OR as well as for ORMR, and thus similar to MR. The AE of the ORMR treatments always outperforms those of OR, but only outperform that of MR depending on N rate, organic class, and its organic N proportion. The MR treatment has the highest AE for a N rate below 100 kg ha⁻¹, but above that MR will gradually be outperformed by classes I, II, and Manure. With higher organic N proportions of ORMR, the AE curves get flatter, but also start lower, and therefore outperform MR only at higher total N inputs. So, while AE of MR was the highest at low N rates, it declined the most rapidly with increasing N inputs, whereas the AE of OR treatments were the lowest at low N rates, but did not decrease substantially with increasing N inputs. By analogy, the same was observed for ORMR with low and high organic proportions, respectively.

## Modeled yield variability

The YIELD and individual organic class models report yield variances, which allow a calculation of yield SD responses relative to predicted yields (Fig 8). The graph indicates that variability of organic input treatments increases with decreasing quality, so in the order of class I, II, Manure, and III. Yield variability for the MR treatment is estimated to be higher than OR class I and II, but lower than OR class Manure and III. Combining the OR treatments with MR, seems to increase the variability of the highest organic quality class I, and decrease the variability of the lowest organic quality class III, without considerable effect on classes II and Manure.

## Modeled SOC

The N-SOC and C-SOC models reveal variances across site being 4.5 and 3.5 times bigger, respectively, than across time, and residual variances smaller than 0.03% (S2 and S3 Tables).

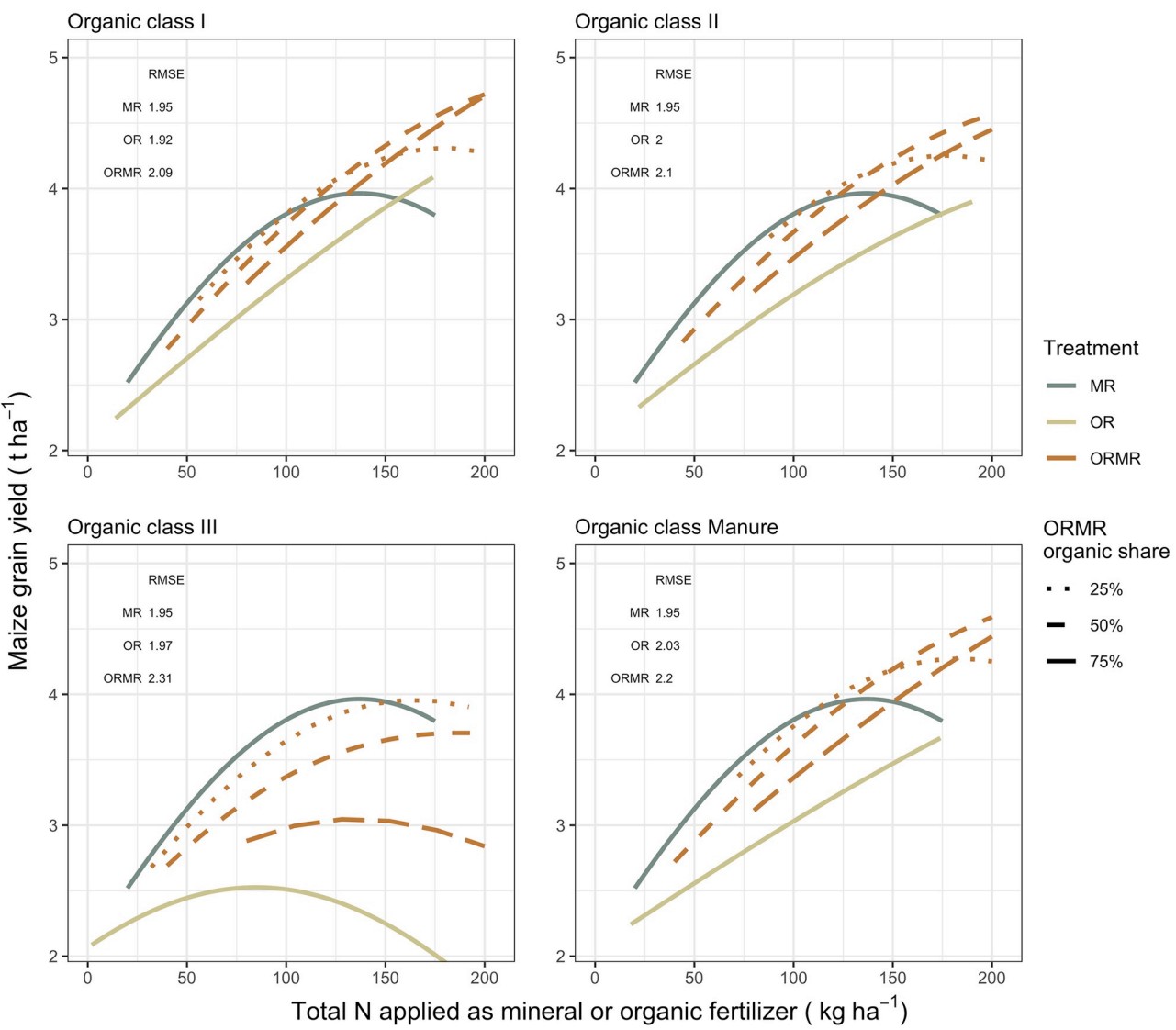

**Fig 6. Model predicted maize grain yield values were plotted for the available total N rate, within a range of 0-200 (kg ha⁻¹), applied as mineral or organic fertilizer, separately for each organic class and ORMR with three different organic N rate proportions.** The root mean square error (RMSE, in t ha⁻¹) is given for the different predicted yields from treatments with the respective organic classes.

Both models also show that neither of the mineral N and organic input interactions are significant, nor are those of mineral N rate and mineral fertilizer design. The N-SOC model estimated that cumulative N rates from all organic quality classes have a significant increasing effect on SOC, but with diminishing responses as seen from the negative quadratic coefficients. These estimates are, however, relatively larger for the low-quality classes III and especially IV. The C-SOC model on the other hand, estimated significant SOC increases for C rates from all classes except for IV, but with relatively greater estimates for high-quality classes. The estimates decrease following the order of class Manure, I, II, III, and IV that was not significant. The initial SOC had a significant negative effect on SOC change, meaning that sites with higher initial SOC on average experience more SOC loss. The predicted dSOC according to N or C inputs are visualized in Figs 9 and 10, respectively.

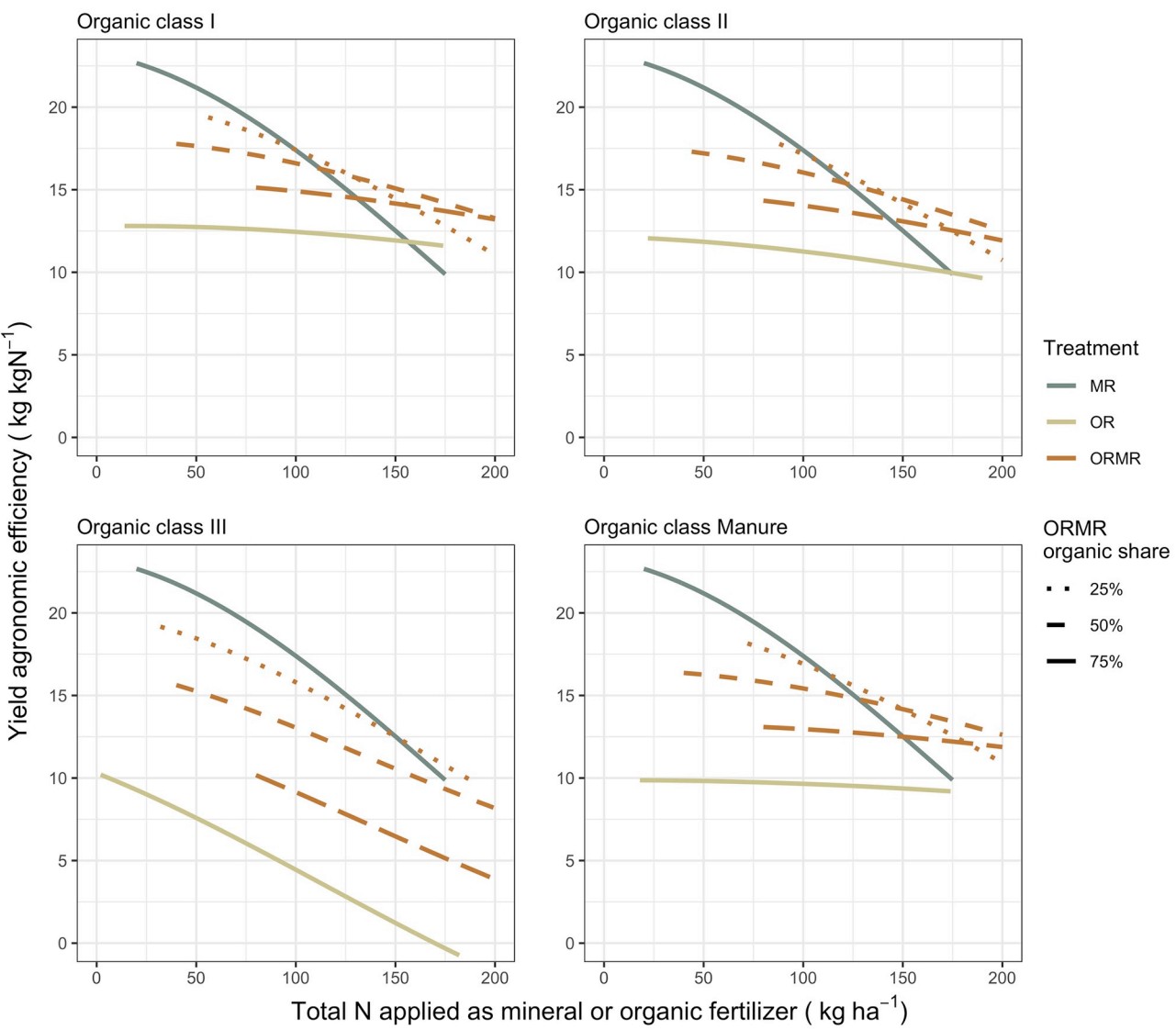

**Fig 7. The agronomic efficiency (AE, in kg kgN$^{-1}$) was plotted for the available total N rate, within a range of 0-200 (kg ha$^{-1}$), applied as mineral or organic fertilizer, separately for each organic class and for ORMR with three different organic N input proportions.**

The N-SOC model predictions of dSOC indicate that the OR treatments have increasingly positive effects on dSOC with increasing N rates, the extent of the effect depending on organic input quality. The positive effects of OR are most pronounced for the high-quality classes I and Manure, followed by decreasing responses from classes II, III, and IV, respectively. The graphs confirm visually that the effect of mineral N is not significant, and nor is its interaction with organic inputs, since the positive effect of ORMR on dSOC seems to be directly related to the organic proportion of ORMR. The C-SOC model predictions of dSOC confirm the model output (Fig 10). On the one hand, organic C inputs from high-quality organic resources have a greater positive effect on dSOC than the low-quality resources, while on the other hand, the effect of mineral N seems negligible and its interaction with organic inputs absent.

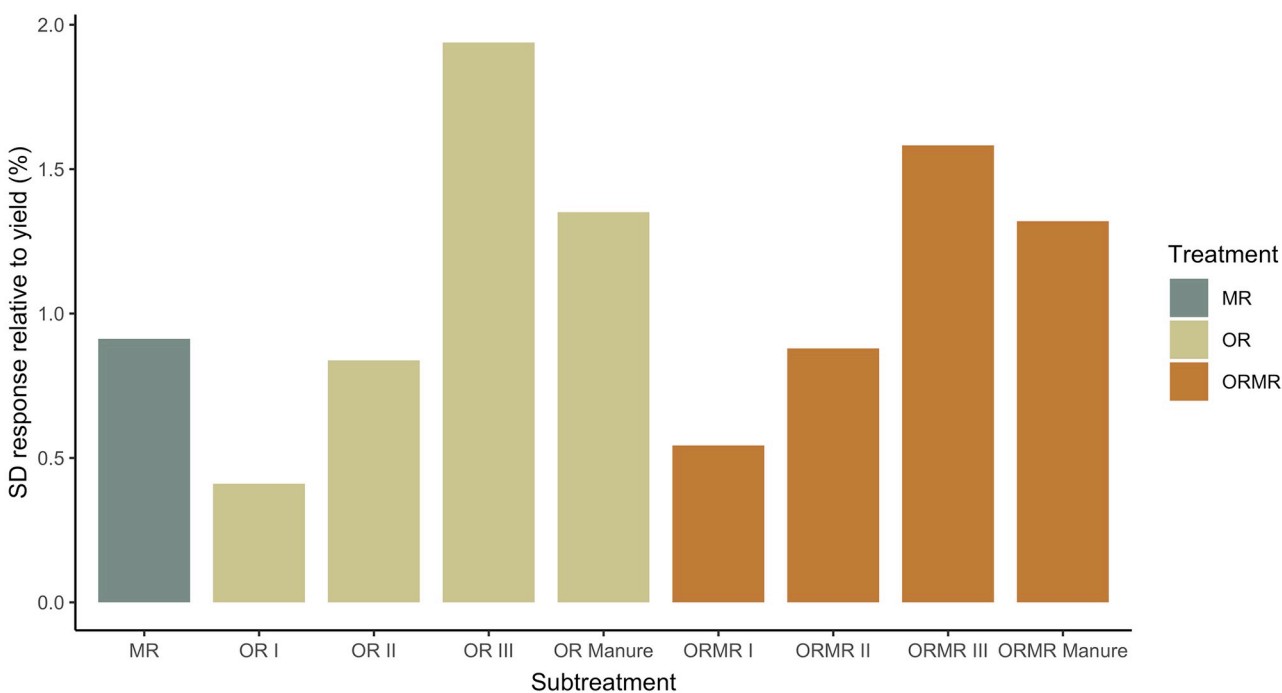

**Fig 8. Yield standard deviation (SD) responses across time are shown for each subtreatment, and relative to their respective predicted yield values.** Values are computed based on the random variances across time from the YIELD and individual organic class models.

## Discussion

The ISFM practice of combined organic and mineral fertilizer application has been found to address the three CSA pillars, but to different extents. These extents depend on input rates and organic resource quality, which determine the combined application effect relative to the sole organic and mineral applications. Our analyses showed that ORMR treatments lead to positive interactions, and to AEs and yields that can outperform those of MR treatments. In fact, the ORMR AE curves are a direct evidence of these positive interactions, as the combination of organic and mineral N resources gives higher yields than the individual resources, and at high N rates. These findings are not entirely lining up with the meta-analysis of Chivenge et al. [23]. Like other studies, they revealed potentially greater yield responses for the combined application treatments [24–27], but could not find evidence for improved AE as predominantly negative interactive effects were observed between the two resources. They found that the AE for OR and ORMR treatments were similar and both lower than the AE of MR treatments, and suggested that potential interactive effects might have been masked with reduced AE due to relatively high total N rates of ORMR. We suspect that the different modeling approach of the current analysis, with in particular the emphasis on N rate effects, might have allowed us to identify positive interactions where Chivenge et al. [23] found none or negative ones. Indeed, including N rates as predictors in the model revealed the importance of looking in detail at N rate for productivity and AE differences between treatments (Figs 6 and 7). In contrast to Chivenge et al. [23], the meta-analysis of Vanlauwe et al. [94] observed that combining mineral N with manure or compost resulted in the highest AE. In our analysis, this observation is more nuanced, as it is true for high total N rates only.

The observed ORMR interactions and their effect on AE and yield, relative to the OR and MR treatments, were affected by both N rate and organic resource quality. The diminishing

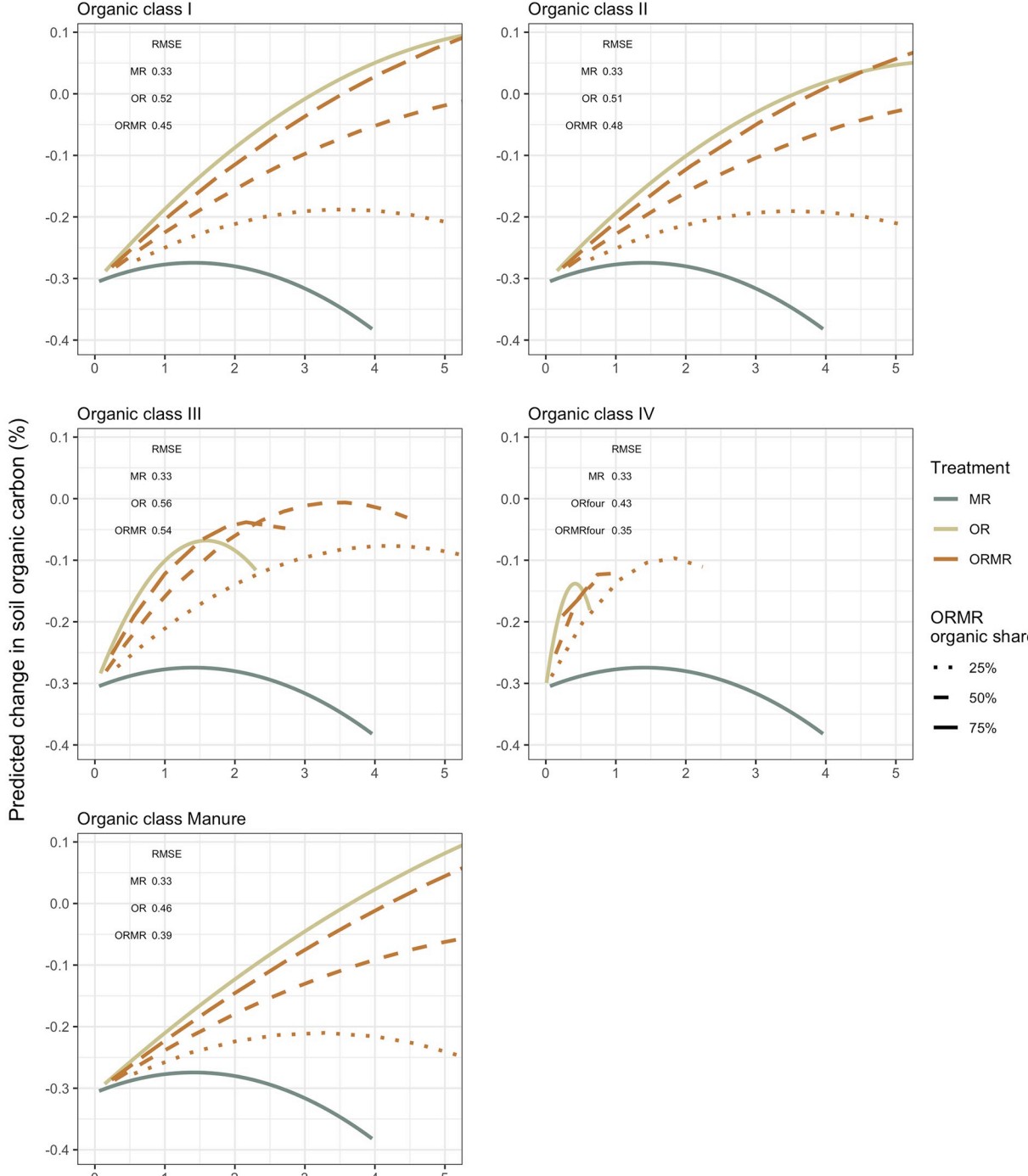

**Fig 9. The output of the N-SOC model is used to predict changes in soil organic carbon from its initial values (dSOC) across the cumulative total N input rates (t ha$^{-1}$), applied as mineral or organic fertilizer and for which data are available, but limited to five cumulative t N ha$^{-1}$.** They are plotted at a data base average initial SOC of 1.8% and separately for each organic class and ORMR with different proportions of organic N content. The root mean square error (RMSE, in %) is given for the different predicted dSOCs from treatments with the respective organic classes.

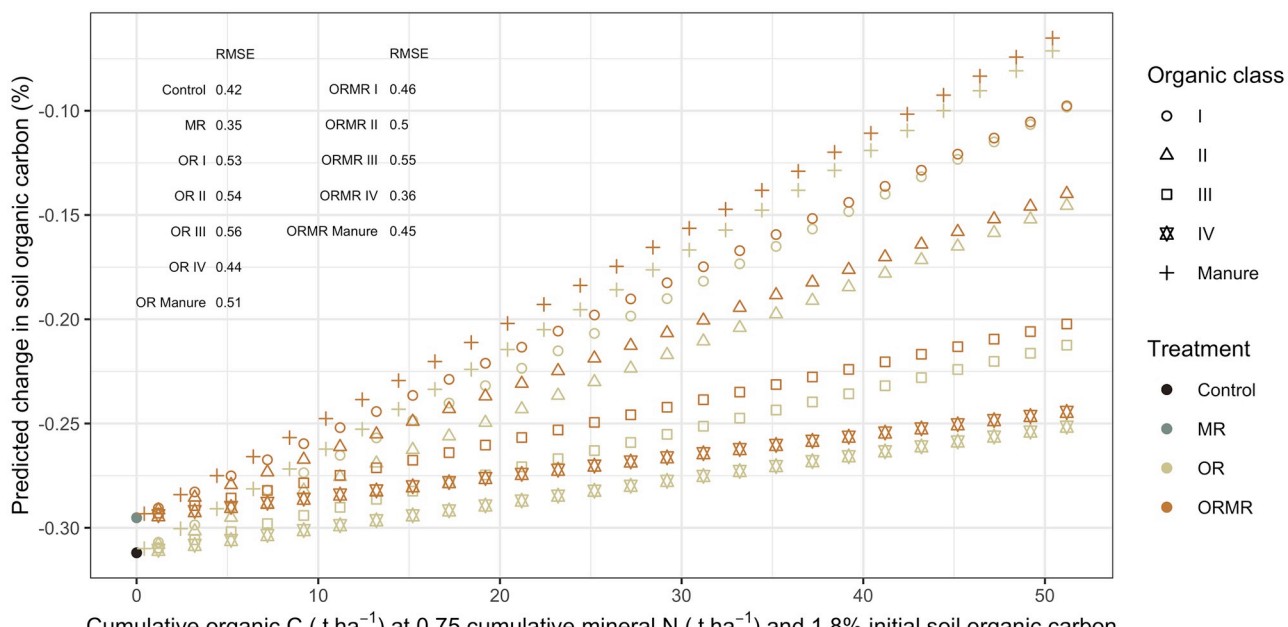

**Fig 10. The output of the C-SOC model is used to predict changes in soil organic carbon from its initial values (dSOC) across cumulative organic carbon input rates.** These are then plotted for each treatment and organic class, and at an initial SOC of 1.8% and cumulative mineral N input of 750 kg ha$^{-1}$. The 750 kg ha$^{-1}$ was based on an annual mineral N application of 75 kg during a time frame of 10 years. The root mean square error (RMSE, in %) is given for the dSOC predictions of the different subtreatments.

yield increases across N rates, as a result of declining AEs across N rates, were apparent for all treatments and hence support earlier findings that the AE decreases with N rate for the organic treatments [23] and MR [94]. The relatively fast AE decline of MR could be explained by an oversupply of N that is exceeding crop uptake, while the more stable AE of organic treatments could be attributed to other soil health benefits than nutrient supply only [103, 104]. The large initial AE of MR and ORMR with low organic N, are likely due to the high plant availability of N when given in its mineral form, which will be lower in applications with higher organic proportions due to possible immobilization [105]. Hence, in our analysis, the ORMR treatments seem to reflect benefits from both sole OR and MR treatments; while their mineral N ensures a relatively high AE, their organic properties contribute to a more stable AE across N input rate.

An important role was observed for organic resource quality on treatment and interaction effects, confirming what has been postulated [28, 47] and observed in previous studies. Our findings agree with those of Chivenge et al. [23], who found greater yield responses for OR and ORMR treatments with high-quality organic resources as opposed to low quality resources. However, our results are not in line with Vanlauwe et al. [94], who found that only OR class II treatments showed significantly higher AE compared to MR treatments, and are in direct contrast with Gentile et al. [24, 25], who found that ORMR treatments with low-quality organic resources tend to favor positive interactions as opposed to high-quality resources. According to the modeled predictions, increasing inputs of low-quality organic resources, such as class III, does not increase yields per se. In fact, low-quality organic resources would increase yields at low rates, but have a detrimental effect at high rates. This can be explained by the fact that low-quality organic resources have a relatively high C:N ratio, which at high input rates can lead to N immobilization in the soil [23, 24]. The immobilization of N consequently lowers the N availability for plants and so affects the yields negatively. The same explanation can also be used for the interpretation of the organic quality effect on AE. Increasing the rate

of low-quality organic resources of class III, likely increases the N-immobilization and would cause the observed sharp declines in AE. However, one would then assume that the N-immobilization of OR class III can be alleviated by mixing it with mineral N, i.e. ORMR class III. While we observed evidence for a slight positive interaction, we would have expected a stronger response based on the given explanation. From our observations, it seems that N immobilization is a more important factor than the potentially enhanced organic N mineralization if high-quality organic resources are combined with mineral N. The latter has been argued by Gentile et al. [24, 25] stating that the excessive amounts of N so supplied would exceed early crop demand, and contribute to negative interactions. In addition, Chivenge et al. [23] suggested that while N immobilization might suppress yields on the short term, this would not necessarily be the case in the long-run due to potential residual effects. However, despite the fact that the meta-analysis included both short- and long-term studies, no evidence was found for the latter.

Many factors like pests and diseases, weeds, and different management practices are involved in causing yield variability over time [106]. Since these factors are assumed to be controlled for in the experiments included in the meta-analysis, we are in fact only considering variability due to meteorological differences between seasons and potential residual effects of treatments. Meteorological differences could be buffered to some extent with soil organic matter build-up, e.g. by improving the soil's water holding capacity [107]. Mineral N on the contrary, has more short-lived effects and potentially even facilitates the degradation of soil organic matter [108]. The long-term trials of Vanlauwe et al. [34] did not show any evidence of the latter, as they observed reduced variability for the ORMR treatments. The suggested rationale was that ORMR has sufficient N from mineral N inputs, while it's organic matter could buffer some of the reduced moisture effects. In this regard we would have expected a lower variability for the OR and ORMR treatments compared to MR. However, our results did not confirm facilitated organic matter degradation by mineral N, nor did they suggest a decreased variability for ORMR compared to OR. Mineral N was not observed to affect the variability of OR treatments when the two resources were combined. Comparing the variability of OR with that of MR indicates that organic resource quality plays a role. The variability of OR was only lower than that of MR for high-quality organic resources of class I and II. Organic class Manure was an exception, as together with the low-quality class III they were estimated to have more variable yields than MR. Chivenge et al. [23] suggested that this organic quality influence could be explained by the greater residual effects observed for high-quality organic resources, whether applied as OR or ORMR. Both our meta-analyses thus challenge the general consensus that a slow decomposition associated with low-quality organic resources would lead to stronger residual effects over time.

Our results confirm observations by Chivenge et al. [23], that SOC can be significantly raised only by adding organic resources to the soil, and that the SOC responses increase with increasing organic N and C input. Mineral N had no significant effect on SOC and conversely there was no interaction with organic inputs. Applying mineral N is expected to enhance root growth and contribute to SOC if the added N can be taken up by the plants, otherwise excess N might facilitate the degradation of soil organic matter with loss of SOC as a result [109]. This could imply an initial increase and consequent decrease of SOC under MR, as was observed by Gentile [109] and Chivenge et al. [23]. The same observation could perhaps be made for the N-SOC model predicted SOC, although the MR response was not significant. The effect of MR on long-term soil organic matter stabilization can be either negative [41] or neutral [38, 42, 50], but generally seems to be complex [108, 109]. The study treatments included in this analysis reported or were assumed to have their above ground crop residues removed after harvest. This implies that an increase of biomass due to an improved soil fertility

could only have benefited SOC through the roots [110]. In a situation where crop residues return completely to the soil, MR (and even Control) treatments are expected to show more positive SOC responses [111].

The effects of organic N and C rate on SOC responses were to a large extent determined by the quality of the organic resource. From the cumulative N input graphs (Fig 9), increasingly steeper response curves were observed for decreasing organic resource quality. The reason is likely that for a certain amount of organic N added, more C will be added by low-quality classes with a relatively high C:N ratio, such as class III and IV. However, for an unbiased view regarding C input, the cumulative C input graphs (Fig 10) show clearly that high-quality organic classes with a relatively high N content and low C:N ratio achieved higher SOC responses than low-quality organic inputs with a high C:N ratio, such as class IV whose responses were not significant. These results are supporting Cotrufo et al. [112] and Castellano et al. [49] who argued that low-quality organic input does not contribute more to soil organic matter (SOM) due to its slower decomposition and consequent accumulation in the soil. Instead, proportionally more SOM would be formed from high-quality organic matter, because a faster decomposition results in more initial microbial biomass, which is increasingly shown to be the largest contributor to stable SOM [112, 113]. Gentile et al. [50], however, found that organic quality did not affect the stabilization of SOC in the long term, and argued that all organic matter would eventually get decomposed and passed through the microbial pool. Perhaps, the reason why this was not observed in the current analysis, is that there were not enough long-term studies included in the analysis that could potentially confirm the arguments of Gentile et al. [50].

We have seen that compared to the sole applications, the combination application has a significant positive effect on maize yield productivity and AE of applied N, which is more pronounced at higher N rates. For yield variability, but especially for SOC, the difference of the combined with the sole organic application was marginal. One can therefore not state that the ISFM combined application decreased yield variability or increased SOC compared to the sole organic and mineral N applications. It has been shown, however, that while the ISFM combined application may not outperform the sole applications on individual CSA pillars, the practice is necessary to simultaneously and optimally address all three pillars. First, mineral N proved to be necessary to achieve the highest productivity and AE for N rates < 100 kg N ha-1. This is important since the global average in 2013 was only 74 kg N ha-1, but it is particularly relevant in the context of smallholder farming in SSA, where mineral N use is even lower yet most yield production increase is needed [114, 115]. Within this N input range, the AE of total applied N would decrease when mineral N is combined with organic inputs, but this is limited when high-quality resources are used. Second, when considering the other two aspects of CSA, we observe that high quality organic inputs become necessary to achieve the lowest yield variability and greatest SOC increase, independent if in combination with mineral N. In summary, the findings of this paper therefore suggest that mineral N is necessary for the best low-input productivity, while high quality organics are necessary for achieving the lowest variability (class I or II) and highest SOC response (class I, II, or Manure). Combining the mineral and organic resources is therefore a way to achieve a win-win-win situation, where productivity, variability, and SOC are all three improved and optimized. For high-input scenarios, the case of combined application is even stronger, as it will outperform the mineral N fertilizer application effects on productivity.

In this paper, organic resources were classified into different organic quality classes, according to their quality parameters N, lignin and polyphenol contents as proposed by Palm et al. [47]. A separate class Manure was made for organic resources like farm yard manure, green manure and compost, whose contents are variable and thus their quality unknown. Yet, based

on N content and productivity and AE response similarities, classes I, II, and Manure, were grouped as high-quality organic resources throughout the analyses. Classes III and IV were consequently grouped as low-quality organic resources. This high- versus low-quality subdivision proved to be relevant, as the effects between high- and low-quality resources on productivity and AE were considerably more distinctive than those within the quality groups. This implies that N content was the main factor contributing to differences in organic resource quality, and confirms that the effect of polyphenol content on productivity and AE is minimal [23]. The high-low quality distinction is less clear for yield variability, due to the relatively high variability under class Manure treatments. For SOC, the distinction is also less pronounced, because the relatively high phenol content of class II compared to class I seemed to have reduced its SOC response. Nevertheless, high-quality organic resources clearly outperform low-quality resources, which challenges the concept of enhanced SOC biochemical stabilization due to the recalcitrance of lignin and polyphenols [47, 50].

An inherent limitation for meta-analyses is the presence of research and publication bias [116]. In this paper, research bias has been addressed by taking variance components as weighting factors in the models run with the *rma.mv* function. Publication bias has been mitigated to an extent by including non-published data, but has also been tested for and evaluated through sensitivity analyses. Another set of limitations to the current meta-analysis are the data constraints that result from the selection criteria. Apart from the main selection criteria such as the specific treatments with maize cropping in SSA, a number of other data variables were often missing because they were either not recorded or simply not reported. Hence, we encourage further research publications to provide yield data for individual seasons instead of averages, variance components of these yield data, input quality parameters, but most importantly organic N rate estimations. The lack of organic N rates alone was accountable for the exclusion of a substantial amount of otherwise useful yield data. For intercropping and rotation systems, especially, the estimation of organic N inputs is challenging, and results in an underrepresentation of these cropping systems in the final data set. Furthermore, it is clear that the majority of studies missed the opportunity to report potential differences in soil characteristics between start and end of experiments, and hence could not be used for the SOC analysis for instance. Similarly, organic quality class IV, and to some extent class III, was underrepresented in the collected literature, and consequently could not take part in the yield analyses. Lastly, we emphasize the need for long-term trials, specifically on weathered soils in the tropics, in order to allow the assessment of treatment effects on yield and SOC over time.

## Conclusion

Meta-analyzing 40 short- and long-term maize nutrient trials across SSA, revealed that compared to sole organic (OR) and mineral N (MR) applications the combined application (ORMR) does have a significant positive effect on maize grain yield productivity and AE of N, albeit more pronounced at higher N rates and for high-quality organic resources. For yield variability and SOC, however, the combined effect is negligible and treatment effects are mainly determined by the organic resource rate and quality. The N rate and organic resource quality were found to play an important role overall, such that compared to low-quality organic resources, increasing amounts of high-quality resources allow for (i) an increased productivity and AE, (ii) a less rapidly declining AE with N rate, (iii) a decreased yield variability, and (iv) an increased SOC.

While the ISFM combined application may not unequivocally outperform the sole applications on individual CSA pillars, the practice has been found to be appropriate for simultaneously and optimally addressing all three pillars and hence can be considered climate-smart.

This paper provides evidence for the need of managing both mineral and organic resources in a holistic and integrated approach, such as aspired by the ISFM framework.

## Supporting information

**S1 Table. Output from the YIELD model.** Run by the *rma.mv* function of the Metafor package in R [117].
(PDF)

**S2 Table. Output from the N-SOC model.** Run by the *lmer* function of the lme4 package in R [99]. Modeled estimates are presented for dSOC, along with p-values for significance [118] and 95% confidence intervals. Extracted and formatted with the stargazer package [119].
(PDF)

**S3 Table. Output from the C-SOC model.** Run by the *lmer* function of the lme4 package in R [99]. Modeled estimates are presented for dSOC, along with p-values for significance [118] and 95% confidence intervals. Extracted and formatted with the stargazer package [119].
(PDF)

**S4 Table. PRISMA checklist.**
(PDF)

## Acknowledgments

The authors would like to thank Pauline Chivenge for sharing her meta-data and giving feedback, acknowledge Monicah Mucheru-Muna, Generose Nziguheba, Geoff Warren, Linus Franke, and Ken Giller for kindly sharing their data as well, Meklit Chernet and Wolfgang Viechtbauer for statistical assistance, and Laurence Jassogne for the initial supervision of this work.

## Author Contributions

**Conceptualization:** Gil Gram, Dries Roobroeck, Pieter Pypers, Johan Six, Bernard Vanlauwe.

**Data curation:** Gil Gram.

**Formal analysis:** Gil Gram, Dries Roobroeck, Pieter Pypers, Johan Six.

**Funding acquisition:** Bernard Vanlauwe.

**Investigation:** Gil Gram, Dries Roobroeck, Pieter Pypers, Johan Six, Bernard Vanlauwe.

**Methodology:** Gil Gram, Dries Roobroeck, Pieter Pypers, Johan Six.

**Project administration:** Gil Gram, Roel Merckx, Bernard Vanlauwe.

**Resources:** Bernard Vanlauwe.

**Software:** Gil Gram.

**Supervision:** Dries Roobroeck, Johan Six, Roel Merckx, Bernard Vanlauwe.

**Validation:** Gil Gram, Pieter Pypers, Johan Six.

**Visualization:** Gil Gram.

**Writing – original draft:** Gil Gram.

**Writing – review & editing:** Gil Gram, Dries Roobroeck, Pieter Pypers, Johan Six, Roel Merckx, Bernard Vanlauwe.

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
