## [Decision Letter · Decision Letter 0]

16 Jul 2020

PONE-D-20-11954

Combining organic and mineral fertilizers as a climate-smart integrated soil fertility management practice in sub-Saharan Africa: a meta-analysis

PLOS ONE

Dear Dr. Gil Gram,

Thank you for submitting your manuscript to PLOS ONE. After careful consideration, we feel that it has merit but does not fully meet PLOS ONE’s publication criteria as it currently stands. Therefore, we invite you to submit a revised version of the manuscript that addresses the points raised during the review process.

We look forward to receiving your revised manuscript.

Kind regards,

Balasubramani Ravindran, Ph.D

Academic Editor

PLOS ONE

Journal Requirements:

Reviewers' comments:

Reviewer's Responses to Questions

**Comments to the Author**

1. Is the manuscript technically sound, and do the data support the conclusions?

Reviewer #1: Yes

Reviewer #2: Yes

2. Has the statistical analysis been performed appropriately and rigorously? 

Reviewer #1: Yes

Reviewer #2: Yes

3. Have the authors made all data underlying the findings in their manuscript fully available?

Reviewer #1: Yes

Reviewer #2: Yes

4. Is the manuscript presented in an intelligible fashion and written in standard English?

Reviewer #1: Yes

Reviewer #2: Yes

5. Review Comments to the Author

Reviewer #1: The manuscript entitled “Combining organic and mineral fertilizers as a climate-smart integrated soil fertility management practice in sub-Saharan Africa: a meta-analysis” presents interesting analysis of various nutrients application for Maize production.

Following are major concerns:

1. What does ‘SE’ represent in the abstract? Put the abbreviation at the correct place in the sentence.

2. The authors need to improve the brevity of the manuscript.

3. The conclusions should be precise and to the point.

4. The use of abbreviations in not consistent throughout the manuscript. For eg. what is MR in line 66? What is ‘AND’ in lines 98 and 99?

5. Expand ‘PRISMA’ in line 103.

6. What exactly do the x and y axis of the graph in figure depict? What are the units and values? Where is statistical analysis for this graph?

7. The presentation of data in form of tables or figures is poor. There are too many figures and tables for the given discussion. Some of the figures can be combined.

8. What is the difference between class I and II, and class III and IV manures? Explain clearly the difference in the composition.

9. What do you mean by dSOC?

10. Based on the results, what kind of treatment would the authors suggest for better yield and productivity in SSA? Please explain in discussion.

Based on the above points, the manuscript may be accepted for publication after ‘Major Revision’.

Reviewer #2: The paper tackles an important issue i.e. Integrated Soil fertility Management in sub-Saharan Africa through combining organic and mineral fertilisers. The authors carried a meta-analysis of short and long-term maize nutrient trials in sub-Saharan Africa so as to evaluate the effect of combined versus sole application of organic and mineral N inputs on (i) maize productivity and Agronomic Efficiency of Nitrogen, (ii) maize yield variability, and (iii) on SOC.

Overall, the manuscript is clearly, concisely and well written. The introduction is relevant and does provide sufficient justification for the review. Moreover, the materials and methods section is also well written. Consistent with the rest of manuscript, the results were also clearly explained and discussed. The figures used, although some not too clear, are relevant and assist in understanding the results better. The conclusions adequately address the objectives set out in the introduction.

Minor Corrections

Line 13: The sentence does not read well, consider deleting the conjunctive adverb "however".

Line 20: it is not particularly clear what "its' refers to in this sentence

Line 23: Add "by' before up to.

Line 268: Replace "about" with of.

Line 290: Replace 'is depending" with "depends"

Line 296: Replace 'is depending" with "depends"

Line 377: Add "of" after the word 'importance"

6. PLOS authors have the option to publish the peer review history of their article (what does this mean?). If published, this will include your full peer review and any attached files.

Reviewer #1: **Yes: **NISHA SHABNAM

Reviewer #2: No

---

## [Author Response · Author response to Decision Letter 0]

20 Aug 2020

Dear editor and reviewers, thank you for considering our manuscript and for the revisions done. We refer to the rebuttal letter where we addressed all of your comments and questions. We remain, however, at your disposal if anything is not clear, or if new questions would arise. We appreciate your work and hope you are satisfied with our answers and revision of the manuscript. Kind regards, Gil (on behalf of the authors)

---

## [Decision Letter · Decision Letter 1]

9 Sep 2020

Combining organic and mineral fertilizers as a climate-smart integrated soil fertility management practice in sub-Saharan Africa: a meta-analysis

PONE-D-20-11954R1

Dear Dr. Gil Gram,

We’re pleased to inform you that your manuscript has been judged scientifically suitable for publication and will be formally accepted for publication once it meets all outstanding technical requirements.

Kind regards,

Balasubramani Ravindran, Ph.D

Academic Editor

PLOS ONE

Reviewers' comments:

Reviewer's Responses to Questions

**Comments to the Author**

1. If the authors have adequately addressed your comments raised in a previous round of review and you feel that this manuscript is now acceptable for publication, you may indicate that here to bypass the “Comments to the Author” section, enter your conflict of interest statement in the “Confidential to Editor” section, and submit your "Accept" recommendation.

Reviewer #1: All comments have been addressed

Reviewer #2: All comments have been addressed

2. Is the manuscript technically sound, and do the data support the conclusions?

Reviewer #1: Yes

Reviewer #2: Yes

3. Has the statistical analysis been performed appropriately and rigorously? 

Reviewer #1: N/A

Reviewer #2: Yes

4. Have the authors made all data underlying the findings in their manuscript fully available?

Reviewer #1: Yes

Reviewer #2: Yes

5. Is the manuscript presented in an intelligible fashion and written in standard English?

Reviewer #1: Yes

Reviewer #2: Yes

6. Review Comments to the Author

Reviewer #1: The authors have responded to all my major concerns in a nice manner. I'm pleased to accept the publication for publication in this journal.

Reviewer #2: (No Response)

7. PLOS authors have the option to publish the peer review history of their article (what does this mean?). If published, this will include your full peer review and any attached files.

Reviewer #1: **Yes: **NISHA SHABNAM

Reviewer #2: No

---

## [Editor Report · Acceptance letter]

14 Sep 2020

PONE-D-20-11954R1 

Combining organic and mineral fertilizers as a climate-smart integrated soil fertility management practice in sub-Saharan Africa: a meta-analysis 

Dear Dr. Gram:

I'm pleased to inform you that your manuscript has been deemed suitable for publication in PLOS ONE. Congratulations! Your manuscript is now with our production department. 

Kind regards, 

on behalf of

Dr. Balasubramani Ravindran 

Academic Editor

PLOS ONE